# Genomic Characterization of *Enterococcus casseliflavus* Isolated from Beef Cows and Calves

**DOI:** 10.3390/microorganisms13040907

**Published:** 2025-04-15

**Authors:** Sani-e-Zehra Zaidi, Rahat Zaheer, Athanasios Zovoilis, Jayce Fossen, Gary Van Domselaar, Cheryl Waldner, Tim A. McAllister

**Affiliations:** 1Department of Biochemistry and Medical Genetics, University of Manitoba, 745 Bannatyne Avenue, Winnipeg, MB R3E 3P5, Canada; sani.zaidi@umanitoba.ca (S.-e.-Z.Z.); athanasios.zovoilis@umanitoba.ca (A.Z.); 2Lethbridge Research and Development Centre, Agriculture and Agri-Food Canada, Lethbridge, AB T1J 4B1, Canada; rahat.zaheer@agr.gc.ca; 3Large Animal Clinical Sciences, Western College of Veterinary Medicine, 52 Campus Dr., University of Saskatchewan, Saskatoon, SK S7N 5B4, Canada; jayce.fossen@usask.ca (J.F.); cheryl.waldner@usask.ca (C.W.); 4National Microbiology Laboratory, Public Health Agency of Canada, Government of Canada, 1015 Arlington Street, Winnipeg, MB R3E 3R2, Canada; gary.vandomselaar@phac-aspc.gc.ca

**Keywords:** antimicrobial resistance, *Enterococcus casseliflavus*, whole genome sequencing, pan-genome, cow–calf system

## Abstract

*Enterococcus* species are used as One Health indicators of antimicrobial resistance (AMR) in humans, animals, and the environment. A surveillance study in beef cows and calves isolated *Enterococcus casseliflavus* along with *E. faecium*, *E. faecalis*, and *E. hirae*. Given the high prevalence of *E. casseliflavus*, we elected to characterize this species to better understand its role in the antimicrobial resistance of enterococci in cows and calves. Almost 12% of *E. casseliflavus* isolates exhibited multidrug resistance with the majority being resistant to lincomycin (99%), followed by quinupristin–dalfopristin (34%), ciprofloxacin (9.6%), tylosin (4.5%), erythromycin (2.7%), tetracycline (1.8%), tigecycline (1.5%), daptomycin (0.6%), streptomycin (0.3%), and kanamycin (0.3%). All *E. casseliflavus* were susceptible to chloramphenicol, penicillin, streptomycin, nitrofurantoin, gentamicin, and linezolid. Whole genome antimicrobial resistance gene profiling identified *vanC*-type intrinsic vancomycin resistance genes in all *E. casseliflavus*, with the *vanC4XYT* gene cluster being dominant (67%) followed by *vanC2XYT* (31%) and *vanC3XYT* (1.5%). Resistance genes for erythromycin (*ermB*) and tetracycline (*tetM*) were rarely identified (2.1% and 1.2%, respectively) within *E. casseliflavus* genomes. No resistance genes were identified to explain either the quinupristin–dalfopristin or ciprofloxacin resistance in these isolates. A core genome phylogenetic tree revealed two clades that exhibited no distinct association with the age of the host, time of sample collection, or the farm sampled. The open nature of the *E. casseliflavus* pan-genome highlighted its intraspecies diversity. These findings suggest that *E. casseliflavus* is likely a low-risk species in terms of contributing to antimicrobial resistance in the cow–calf sector.

## 1. Introduction

The prevalence of antimicrobial resistance (AMR) in the agriculture sector potentially represents both a food safety and public health concern [1,2]. In North America, the beef production system consists of cow–calf operations, backgrounding and finishing feedlots, and meat processing plants. Most cow–calf producers raise cow–calf pairs on pasture, with the calves being weaned and sold to feedlots in the fall. Consequently, the cow–calf sector relies primarily on forages as feed, whereas feedlots include more cereal grain within total mixed diets. Most genomic studies on AMR have focused on more intensively managed swine, dairy, and feedlot cattle, with comparatively few studies of the less intensively managed cow–calf sector.

*Enterococcus* spp. are abundant in the gastrointestinal tract of warm-blooded animals, including humans and livestock. Excreta from these hosts enable enterococci to enter soil, surface, and groundwater, which can serve as indicators of faecal contamination [3,4]. The widespread presence of enterococci in humans, animals, and the environment also makes them useful One Health indicators of AMR [5,6]. Enterococci can also thrive in harsh environments owing to their resistance to disinfectants, intrinsic antimicrobial resistance, and genome plasticity [7].

Enterococci have shown remarkable adaptability, acquiring resistance to nearly all clinically employed antimicrobials. For example, resistance to chloramphenicol, erythromycin, and tetracyclines arose soon after these antimicrobials were available for empirical treatment in clinical settings [3]. This adaptability highlights the critical need for ongoing surveillance and the development of novel strategies for the effective management of enterococcal infections. For this purpose, *Enterococcus* spp. can be used to study the correlation between the prevalence of antimicrobial resistance genes (ARGs) and AMR from a One Health perspective. For instance, in beef production systems, *Enterococcus* species have been used as indicators of the prevalence of macrolide and tetracycline resistance as both antimicrobial classes are used both prophylactically and therapeutically in beef cattle.

Both *E. faecium* and *E. faecalis* are often associated with nosocomial infections in immunocompromised individuals [8]. Occasionally, other species like *E. casseliflavus*, *E. hirae*, *E. avium*, *E. durans*, *E. gallinarum*, and *E. raffinosus* have also been implicated in opportunistic infections in humans [9]. There is even evidence that this host–species relationship may be specific to the stage of beef cattle production, with *E. hirae* being predominant in feedlot cattle [5,10,11] and *E. casseliflavus* in cows and calves [12]. Still, the drivers that promote *E. casseliflavus* in the cow–calf sector are unknown. This study addresses this gap by conducting a phenotypic and genotypic characterization of *E. casseliflavus* isolated from cows and calves.

## 2. Methodology

### 2.1. Sample Collection and Processing

This genomic study is based on 331 *E. casseliflavus* isolates recovered from faecal samples collected from 39 cow–calf herds in the spring and fall of 2010 across four provinces: Alberta (*n* = 20, 51%), Saskatchewan (*n* = 9, 23%), British Columbia (*n* = 5, 13%), and Manitoba (*n* = 5, 13%); [12]. This project was approved by the University of Saskatchewan’s Animal Research Ethics Board (AREB) under the animal use protocol # 2014003 and complies with Canadian Council on Animal Care (CCAC) guidelines, the University of Saskatchewan Animal Care and Use Procedures, and the Tri-Council MOU—Schedule 3: Ethical Review of Research Involving Animals.

*Enterococcus* species isolates were recovered from pre-enriched faecal samples (incubated in 1% buffered peptone water for 1 h) onto mEnterococcus agar as described previously [12]. Briefly, presumptive *Enterococcus* colonies were sub-cultured on Columbia agar with 5% sheep blood. Isolates from colonies were initially identified as *Enterococcus* using matrix-assisted laser desorption ionization time-of-flight mass spectrometry (MALDI-TOF MS; Bruker Daltonik, Bremen, Germany; [13]).

### 2.2. Whole Genome Sequencing

The identity of the 331 *E. casseliflavus* isolates reported in this analysis was confirmed using whole genome short-read sequencing using an Illumina NovaSeq 6000, generating 2 × 250 base-paired end reads. Genomic DNA from bacterial isolates was extracted using an Applied Biosystems Magmax DNA Multi-Sample Ultra 2.0 kit (Thermo Fisher Scientific, Waltham, MA, USA) with MagMAX™ Cell and tissue DNA extraction buffer (Thermo Fisher Scientific) in a KingFisher™ Flex 96 deep-well magnetic particle processor (Thermo Fisher Scientific). Wash plates (wash I solution: 1000 μL per sample; wash II, plate 1: 1000 μL of 80% ethanol per sample; wash II, plate 2: 500 μL of 80% ethanol per sample) and DNA binding bead mix (400 μL Lysis binding mix + 40 μL of binding beads per sample) were prepared as per the manufacturer’s protocol. Samples were eluted in 300 μL of 10 mM Tris-HCl PH 8.0 +0.1 mM EDTA. For each isolate, a 10 μL loop of culture was suspended in a 1.5 mL microcentrifuge tube with 200 μL of a Gram-positive cocktail mix [175 μL of PBS PH:7.4, 20 μL 100 mg/mL lysozyme; (Sigma Aldrich, St. Louis, MO, USA), 5 μL 25 KU/mL Mutanolysin (Sigma Aldrich)]. The suspended cells were then incubated at 37 °C for 90 min. Tubes were centrifuged, and the pellet was resuspended in 560 μL of lysis mix [500 μL MagMAX™ Cell and Tissue DNA extraction buffer, 20 μL enhancer solution, 40 μL proteinase K, 10 μL PureLink RNase A, (Invitrogen, Waltham, MA, USA)] and mixed by repeatedly pipetting until there were no visible particles. The lysate was transferred to a Kingfisher 96 deep-well plate, and 440 μL of DNA binding bead mix was added to each well with lysate and mixed on a plate shaker at 400 rpm for 5 min. The MMX_Ultra2_Cell_Tissue_96_V2_Flex program was selected in the Pharma KingFisher™ Flex 96 Deep-Well Magnetic Particle Processor (Applied Biosystems™, Waltham, MA, USA), and the plates were loaded as per program specifications. After the completion of the program, elution plates were sealed using Thermo Fisher Scientific sealing tape and incubated at 65 °C for 15 min, followed by mixing on a plate shaker for 15 min. After centrifugation, the elution plate was placed on a magnetic rack to separate the residual magnetic beads left over from the extracted DNA. Once the beads were separated, 100 μL of eluate was transferred to an Eppendorf twin.tec 96-well plate full skirt (Eppendorf, Mississauga, ON, Canada) and sealed using Life Technologies MicroAmp^®^ clear adhesive film (Thermo Fisher Scientific). The remaining DNA was aliquoted into labelled 1.5 mL centrifuge tubes for storage. The eluted DNA from the plate was then quantified using a Qubit fluorometer, and the A260/280 and A260/230 ratios were measured using a Nanodrop spectrophotometer to ensure that quality metrics (DNA yield < 150 ng/μL, A260/280 = 1.8–2.0, A260/230 = 2.0) for sequencing were met. Barcoded sequencing libraries were prepared for each genomic DNA and sequenced on an Illumina NovaSeq 6000 SP platform PE250 to generate 2 × 250 bp paired-end reads. All genomic sequencing was performed at Genome Quebec sequencing services (Montreal, PQ, Canada).

### 2.3. Genomic Analysis of Enterococcus casseliflavus

De-multiplexed sequence reads were de novo-assembled using the Unicycler pipeline v0.4.8.0 (https://github.com/rrwick/Unicycler, accessed 10 July 2024) to generate bacterial genomes. Illumina adapters were removed from reads using Trimmomatic v0.36.5. Genome assemblies were evaluated by QUAST v5.0.2, followed by gene annotation using Prokka v1.14.6 [13].

Assembled genomes were tested for the presence of antimicrobial resistance genes, plasmids, and sequence types using the Staramr tool [14]. All genomes were also screened for the presence of virulence genes against VirulenceFinder database [15] using ABRicate tool v 0.9.8 [16].

Comparative genomic analysis was conducted using the Roary v3.12.0 pipeline with default parameters [17]. All assembled *E. casseliflavus* genomes (*n* = 331) were subjected to phylogenomic analysis. A core genome-based neighbour-joining phylogenomic tree was constructed using core gene alignment from Roary with Geneious Tree Builder v10.2.6. The generated phylogenomic tree and associated metadata were visualized using the Interactive Tree Of Life (iTOL) v 6 tool [18]. The pan-genome composed of 331 *E. casseliflavus* isolates was reconstructed and annotated for core and accessory genes using Prokka. The phandango interactive viewer tool was used to visualize pan-genome data generated from Roary [19]. A gene absence and presence matrix file was utilized to create a heat map based on the number of genes present or absent in each isolate, and a Newick-formatted tree file of accessory genomes generated by Roary was used to plot a relatedness dendrogram of the accessory genes present in all isolates.

A pan-genome plot was generated using the ggplot2 package of R Studio (v 2024.09.0 + 375) (R Studio Inc., Boston, MA, USA). The plot was constructed using Roary to generate output files estimating the number of conserved and total genes. The number of conserved genes determined the size of the core genome in a dataset. The number of total genes in a dataset consisted of both the core and accessory genomes, creating a curve that described the pan-genome. Pan-genome size was calculated using the Heap law equation, n = κNγ [20]. Additionally, to characterize the accessory genome, a plot was constructed to describe the genes associated with clusters of isolates as well as genes that were unique to a specific isolate.

### 2.4. Antimicrobial Susceptibility Testing

With the objective of identifying correlations between AMR genotype and phenotype, the 331 *E. casseliflavus* isolates were tested for phenotypic resistance using broth microdilution. Minimal inhibitory concentrations (MICs) were determined according to Clinical and Laboratory Standards Institute (CLSI) guidelines using NARMS CMV3AGPF Sensititre plates recommended for Gram-positive bacteria (Thermo Fisher Scientific, Waltham, MA, USA). All isolates were tested for susceptibility to 16 antimicrobials with serial dilutions across the following specified concentration ranges (μg/mL): chloramphenicol (2–32), ciprofloxacin (0.12–4), daptomycin (0.25–16), erythromycin (0.25–8), gentamicin (128–1024), kanamycin (128–1024), lincomycin (1–8), linezolid (0.5–8), nitrofurantoin (2–64), penicillin (0.25–16), quinupristin–dalfopristin (0.5–32), streptomycin (512–2048), tetracycline (l–32), tigecycline (0.015–0.5), tylosin (0.25–32), and vancomycin (8–32). Detailed methods for susceptibility testing have previously been reported [12].

## 3. Results

### 3.1. Recovery of Enterococcus casseliflavus

*E. casseliflavus* was consistently recovered as one of the prominent species in cows and calves sampled in the fall and spring (Figure 1; [12]). This species accounted for 42%, 36%, and 37% of the isolates from spring cows, fall calves, and fall cows. *E. casseliflavus* was the second most frequently recovered species from spring calves (23%), with *E. faecalis* (24%) being slightly more prevalent.

### 3.2. Genomic Characterization of E. casseliflavus

*E. casseliflavus* genomes ranged from 3,384,635 bp to 4,082,546 bp, with an average GC content of 42.4% (Table 1). The vancomycin *vanC*-type locus (*vanC-vanXY-vanT*) was present in all *E. casseliflavus* genomes, with *vanC4XYT* found in 67% (222/331) of the genomes. Other *vanC* variants, including *vanC2XYT* (31.4%, 104/331) and *vanC3XYT* (1.5%, 5/331), were also identified. All *vanC* resistance determinants were upstream from *vanR* and *vanS*, constituents of the two-component regulatory system for vancomycin resistance. The erythromycin resistance gene, *ermB*, was found in only 7 of 331 (2.1%) *E. casseliflavus* genomes. Mutations in the 23S RNA gene conferring erythromycin resistance were not found. Additionally, no known mutations in the *parC* or *gyrA* genes linked to ciprofloxacin resistance were identified in any of the *E. casseliflavus* genomes. Furthermore, no genes associated with quinupristin–dalfopristin, such as *lsa* [21] or *vatE* [22], were identified. The screening of genomes against the Virulence Finder database (VFDB) identified the ecbA/fss3 adhesion gene in 9% (30/331) of the *E. casseliflavus* genomes.

Plasmid profiling identified the presence of rep1, repUS1, repUS7, and repUS58 plasmids [23]. The repUS1 plasmid was found in 23% of genomes (77/331) from cows and calves in both seasons. In contrast, rep1 and repUS7 were found only once in the genome of an isolate recovered from a fall calf, with the repUS58 plasmid being associated with the genome of a single isolate obtained from a spring calf. No antimicrobial resistance genes were found on any of the identified plasmids.

The core genome-based phylogenomic tree formed two major clades (Figure 2). No segregation/clustering was observed based on the season or farm sampled or whether they originated from cows or calves. The nature of the *E. casseliflavus* pan-genome was assessed using Heap’s law (n = κN^γ^; [20]), where the model estimates the number of new gene clusters observed as genomes are randomly ordered. The decay parameter γ determines the openness of the pan-genome: if γ > 1.0, the pan-genome is considered closed, whereas if γ < 1.0, it is open. The analysis revealed an open pan-genome with a γ value of 0.3. Further analysis with Roary identified 1963 core genes (99 to 100% of strains), 184 soft core genes (95 to 99% of strains), 2243 shell genes (15 to 95% of strains), and 18,626 cloud genes (0 to 15% of strains) (Figure 3). The phylogenetic tree showed two distinct clusters, but this distinction did not appear to be related to the time, source or location of isolate collection (Figure 4). 

### 3.3. Phenotypic Resistance Profiling

A total of 331 *E. casseliflavus* isolates were tested for antimicrobial susceptibility against 16 different antimicrobials to investigate the correlation between phenotypic and genotypic AMR profiles (Figure 5). Across all isolates, 12% were multidrug-resistant (resistant to ≥3 antimicrobials). Most isolates were resistant to lincomycin (99%, 328/331), followed by quinupristin–dalfopristin (34%, 114/331), ciprofloxacin (9.6%, 32/331), tylosin (4.5%, 15/331), erythromycin (2.7%, 9/331), tetracycline (1.8%, 6/331), tigecycline (1.5%, 5/331), daptomycin (0.6%, 2/331), streptomycin (0.3%, 1/331), and kanamycin (0.3%, 1/331). None of the isolates exhibited resistance to chloramphenicol, penicillin, nitrofurantoin, gentamicin, or linezolid. Most isolates appeared susceptible to vancomycin with MIC values of ≤2 µg/mL (7.5%) and 4 µg/mL (65.5%), while 27% of the isolates exhibited intermediate resistance (MIC: 8 µg/mL).

Ciprofloxacin-resistant *E. casseliflavus* were more prevalent in cows (13.1%) and calves (17.0%) in the fall than the spring (3.2%—cows; 1.8%—calves). Daptomycin resistance was only found in a single isolate from spring cows (1.0%, 1/93) and a single isolate from fall cows (1.0%, 1/91). Likewise, streptomycin- and kanamycin-resistant isolates were rare in spring cows and calves (streptomycin, cows, 1.0%, 1/93; kanamycin calves 1.8%, 1/53). There were no differences between quinupristin–dalfopristin resistance in isolates collected from cows and calves or collected in the spring and fall.

## 4. Discussion

Members of the genus *Enterococcus* have been extensively used as indicator species of faecal contamination and in antimicrobial resistance surveillance studies [24,25,26]. Despite the ubiquitous presence of *Enterococcus* spp. across all sectors, including humans, animals, and the environment, previous studies have indicated niche specificity among *Enterococcus* species. Among all *Enterococcus* species, *E. faecium* and *E. faecalis* are frequently associated with nosocomial infections in humans [6,8,27], as well as being members of the normal gut microbiota of humans [28]. In contrast, *E. hirae* exists predominantly in feedlot cattle and swine [6,11]. Previous studies collecting isolates from cows and calves showed an increased prevalence of *E. casseliflavus* compared to feedlot cattle [12].

Originally isolated from a variety of plants [29], *E. casseliflavus* has since been recovered from animal faeces and environmental samples collected in multiple studies [30]. *E. casseliflavus* demonstrates unique ecological and functional roles, such as degrading organic pollutants like decabromodiphenyl ether (BDE-209), a common brominated flame retardant [31], and mediating the detoxification of the insecticide chlorantraniliprole in the agricultural pest *Spodoptera frugiperda* [32]. In aquaculture, *E. casseliflavus* acts as a probiotic in rainbow trout, enhancing gut health and disease resistance through immunomodulation [33]. Despite its predominant role as a plant and animal commensal, there are some reports of *E. casseliflavus* causing clinical infections in humans [10]. Compared to clinically significant *Enterococcus* spp. like *E. faecium* and *E. faecalis*, the prevalence of *E. casseliflavus* in clinical settings (>1.3% cases) is exceedingly low [34] and has only been rarely associated with bacteremia in humans [35,36]. In dairies, *E. casseliflavus* has been associated with subclinical mastitis in sheep and goats [37]. Despite the importance of *E. casseliflavus* from a One Health perspective, knowledge of its genomic structure remains limited. This study aims to bridge this gap.

Surveillance studies conducted in cow–calf herds in both Canada and the United States found an increased prevalence of *E. casseliflavus* compared to feedlot systems [12]. In contrast to intensive grain-based feedlot cattle production, cow–calf production uses more forages, where animals depend on grazing supplemented with hay and silage as the primary sources of feed. Since *E. casseliflavus* is known as a plant-associated enterococcal species, the higher prevalence of *E. casseliflavus* in cow–calf systems may be a reflection of frequent inoculation with *E. casseliflavus* from forages. In feedlot cattle, *E. hirae* predominates owing to the selective pressures associated with the feeding of grain-based diets [11].

There is a notable difference in antimicrobial usage in cow–calf vs. feedlot production systems. In cow–calf operations, the overall frequency of antimicrobial use is comparatively low, and antimicrobials are primarily used to treat disease in individual animals [12], whereas in feedlots, antimicrobials are frequently administered to prevent disease in the herd, either via injection or inclusion in feed [12,38,39]. In surveillance studies, enterococci serve as an indicator of antimicrobial resistance, and a previously high prevalence of tetracycline and macrolide resistance genes was found in *Enterococcus hirae*, the dominant species in feedlot cattle [6,11,40]. In contrast, the tetracycline resistance gene *tetM* and the macrolide resistance gene *ermB* were rarely detected in *E. casseliflavus*. This suggests that the low prevalence of tetracycline and macrolide resistance in cows and calves is a reflection of the limited use of these antimicrobials in this sector. Furthermore, it also strengthens the hypothesis that the prophylactic and metaphylactic use of tetracyclines and macrolides in feedlots, along with other environmental factors, promotes the prevalence of *E. hirae* in feedlot cattle. *E. hirae* has likely thrived in feedlot systems due to its acquisition of resistance to the antimicrobials commonly used in feedlot cattle. Moreover, *E. hirae* appears to have unique genes associated with vitamin production, cellulose, and pectin degradation, traits which may also support its adaptation to the bovine digestive tract.

*Enterococcus casseliflavus* typically exhibits low-level vancomycin resistance, with MICs ranging from 2 to 32 µg/mL, while remaining susceptible to teicoplanin (≤1 mg/L) [9,41]. This resistance is attributed to the intrinsic presence of the chromosomally encoded *vanC* gene cluster, a trait that is also present in *E. flavescens* and *E. gallinarum* [42]. The *vanC* cluster comprises five genes, *vanC*, *vanXYc*, *vanT*, *vanR*, and *vanS* [43], with *vanC*, *vanXYc*, and *vanT* being essential for resistance. VanC is a ligase that synthesizes d-Ala-d-Ser for alternative cell wall precursors, VanT supplies d-Ser as a membrane-bound serine racemase, and VanXYc combines the functions of the VanX and VanY enzymes to cleave d-Ala-d-Ala precursors. The remaining genes, *vanR* and *vanS*, form a two-component regulatory system. Unlike high-level glycopeptide resistance phenotypes (VanA, VanB, and VanD), which utilize d-Lac precursors, the VanC, VanE, and VanG phenotypes result from the replacement of d-Ala with d-Ser [44]. Although the resistance level conferred by the *vanC* cluster is lower than that of other vancomycin-resistant enterococci (VRE), this resistance could limit treatment options and complicate clinical management should *E. casseliflavus* play a more prominent role in nosocomial infections [7]. Due to this moderate resistance, the Clinical and Laboratory Standards Institute (CLSI) classifies *Enterococcus* species with vancomycin MICs below 4 mg/L as susceptible. However, vancomycin is not actively recommended for the treatment of *E. casseliflavus* bacteremia [9]. The accurate identification of *E. casseliflavus* is thus critically important for effective disease treatment, and misidentification can lead to inappropriate therapeutic strategies and potentially poor patient outcomes. From an AMU perspective, the misuse of vancomycin for treating *E. casseliflavus* infections may also expose other bacterial species to vancomycin, potentially resulting in the proliferation of vancomycin-resistant strains as result of selective pressure [35]. Variants of the *vanC* cluster [45,46] were found in all *E. casseliflavus* genomes in this study and harboured all five genes, *vanC*, *vanXY*, *vanT*, *vanR*, and *vanS*. Compared to *vanA* and *vanB*, which are often associated with mobile genetic elements, the *vanC* cluster was chromosomally encoded. This likely limits the transfer potential of *vanC* to other enterococcal species [47], an outcome that may contribute to the rarity or absence of vancomycin resistance in enterococci species isolated from feedlot cattle [6,11].

Enterococci are intrinsically tolerant to all beta-lactams and resistant to clinically achievable levels of aminoglycosides, lincosamides, and trimethoprim–sulfamethoxazole [21]. *E. casseliflavus* is a member of the *E. gallinarum* group, which exhibit intrinsic resistance to several antimicrobials. Nearly 98% of *E. casseliflavus* isolates were phenotypically resistant to lincomycin. Previously, lincomycin resistance was observed in 81% of *E. casseliflavus* isolates recovered from vegetables [48].

Virginiamycin, a streptogramin class of antibiotics, has been extensively used in human medicine and livestock, primarily as a growth promoter and to prevent liver abscesses. However, its use was discontinued in Europe in 1999 due to concerns over cross-resistance against the medically important antimicrobial combination quinupristin–dalfopristin (Q/D), used to treat vancomycin- and linezolid-resistant *E. faecium* infections [49,50]. A reduction in the use of Q/D in humans has led to a resurgence of the use of virginiamycin in livestock. Our study identified Q/D resistance in 34% of *Enterococcus casseliflavus* isolates, consistent with the findings from CIPARS, which reported a significant rise in streptogramin resistance among *Enterococcus* species, increasing from 15% to 35% since 2019. However, virginiamycin is not labelled for use in cow–calf herds as liver abscesses are not a common issue with grazing or high-forage-based diets, and its use was not reported in the herds participating in the present study [12]. Resistance to streptogramins is associated with antibiotic inactivation mediated by acetyltransferase encoded by *vatD*, *vatE*, and *vgbA* [51,52]. Despite the recovery of Q/D-resistant *E. casseliflavus*, no genes previously associated with Q/D resistance were detected in the *E. casseliflavus* genomes, raising the possibility that unknown ARGs or mutation(s) may be conferring this resistance. Further research to define the genetic factors responsible for Q/D resistance in *E. casseliflavus* is warranted.

Ciprofloxacin is a broad-spectrum fluoroquinolone that inhibits topoisomerases and thereby prevents DNA replication. In enterococci, ciprofloxacin resistance is associated with parC and gyrA gene mutations. In a previous study investigating ciprofloxacin resistance in *E. casseliflavus*, isolates with mutations in *parC* and *gyrA* did not exhibit phenotypic resistance. However, mutations in codon 80 in ParC (Ser to Leu) and in codon 87 in GyrA (Glu to Gly) resulted in *E. casseliflavus* isolates exceeding the MIC cut-off value (>8 µg/mL) for ciprofloxacin resistance [53]. In this study, ciprofloxacin resistance isolates had an MIC <8 µg/mL, and mutations in *parC* or *gyrA* were not identified. This agrees with previous observations that *E. casseliflavus* is less sensitive to ciprofloxacin as compared to other enterococcal species [53]. Unexplainably, we did find that *E. casseliflavus* recovered from cows and calves in the fall exhibited greater ciprofloxacin resistance than those isolated in the spring. This result was especially surprising given the infrequent use of fluoroquinolones within cow–calf herds [38].

Virulence factors in *Enterococcus* spp., apart from the clinically significant *E. faecium* and *E. faecalis*, are generally rare or absent. However, in *E. casseliflavus*, the fibrinogen-binding MSCRAMM gene (*fss3*) was identified. This gene belongs to a family encoding Microbial Surface Components Recognizing Adhesive Matrix Molecules (MSCRAMMs), which play a crucial role in adhesion, biofilm formation, and pathogenicity in enterococci. Previously, *fss3* was also detected in *E. hirae* from bovine faeces and *E. lacertideformus* from wild Asian house geckos [54]. Surveillance studies have reported the presence of *fss3* in clinical *E. faecium* and *E. faecalis* isolates but not in those isolated from beef cattle [6]. Comparative analyses suggests that virulence factors including *fss3* are part of evolutionary adaptations that enable *Enterococcus* species to thrive in diverse environments and hosts [54,55].

Like other enterococcal species, the pan-genome of *E. casseliflavus* was open [56], with the accessory genome accounting for 81% of the genes in the pan-genome. A plot of unique versus new gene families showed that unique gene families are increasing within the pan-genome compared to new families. This suggests that *E. casseliflavus* possesses a diverse range of gene families. Its larger genome size (~3.7 Mb) compared to the opportunistic human pathogens *E. faecalis* (~2.7 Mb) and *E. faecium* (~2.7 Mb) and cattle-associated species *E. hirae* (~2.9 Mb) may be attributed to its environmental versatility and motility-related genes. *E. casseliflavus* is often found in diverse habitats such as plants, soil, and water, which likely requires a broader genetic repertoire to adapt to these environments. This broader adaptability involves additional genes for survival, motility, and metabolic versatility, reflecting its environmental and evolutionary distinctions [57]. The core genome-based phylogenetic tree formed two main clades with no segregation based on animal type, seasons, or farm location. The selective factors that contribute to the separation of these two clades remain elusive.

To summarize, *E. casseliflavus* is a prominent enterococcal species within the cow–calf sector, exhibiting notable genomic diversity among isolates. Consequently, it may be a more reliable indicator of AMR in cow herds than *E. hirae*, which tends to be the predominant species in feedlot cattle. Both phenotypic and genomic characterization suggest low-level resistance to key antimicrobials, including tetracycline and macrolides, which are routinely used in both human and animal medicine. While phenotypic studies identified moderate levels of resistance to quinupristin–dalfopristin in isolates from cows and calves across all time points and higher than expected levels of ciprofloxacin resistance in isolates from cows and calves in the fall, no genomic determinants were identified to account for these findings. Vancomycin resistance in *E. casseliflavus* was associated with a chromosomally located intrinsic resistance gene cluster, suggesting a limited role for this species in the spread of vancomycin resistance among enterococci in this sector.

## Figures and Tables

**Figure 1 microorganisms-13-00907-f001:**
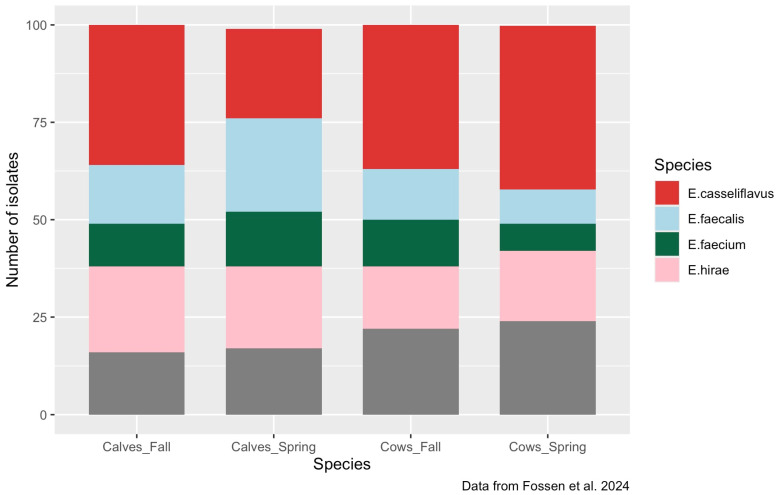
The prevalence of *Enterococcus* species isolated from 1505 cows and calves (743 cows, 762 calves) originating from 39 herds in the spring and fall of 2021 [12].

**Figure 2 microorganisms-13-00907-f002:**
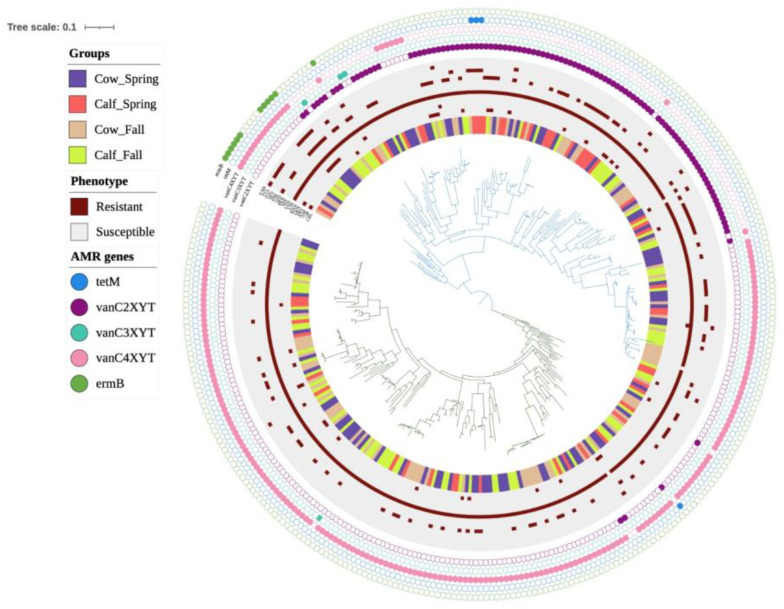
Core genome neighbour-joining phylogenomic tree of *Enterococcus casseliflavus* genomes (*n* = 331) isolated from cows and calves in spring or fall. *E. casseliflavus* EC20 (GenBank accession # CP004856.1) was used as reference genome.

**Figure 3 microorganisms-13-00907-f003:**
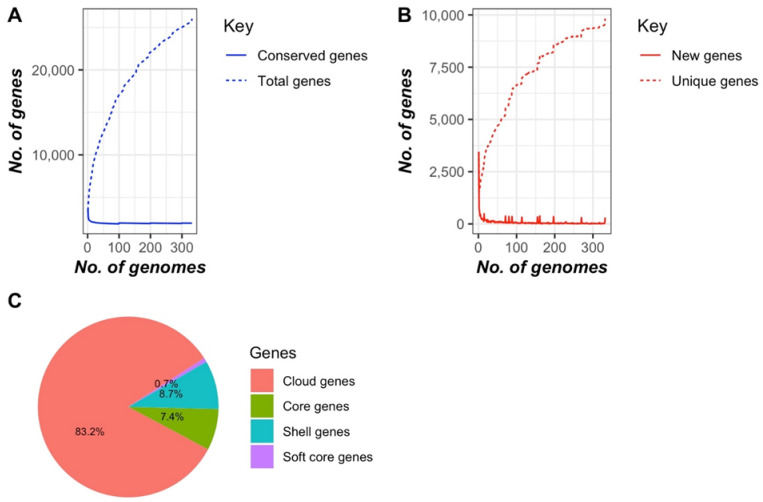
Pan-genome of *Enterococcus casseliflavus* (*n* = 331) (**A**) Conserved and total genes, (**B**) New and unique genes and (**C**) Overall pan-genome, Development of pan- and core genomes was based on conserved genes and presence of new/unique genes, illustrating the open nature of pan-genome.

**Figure 4 microorganisms-13-00907-f004:**
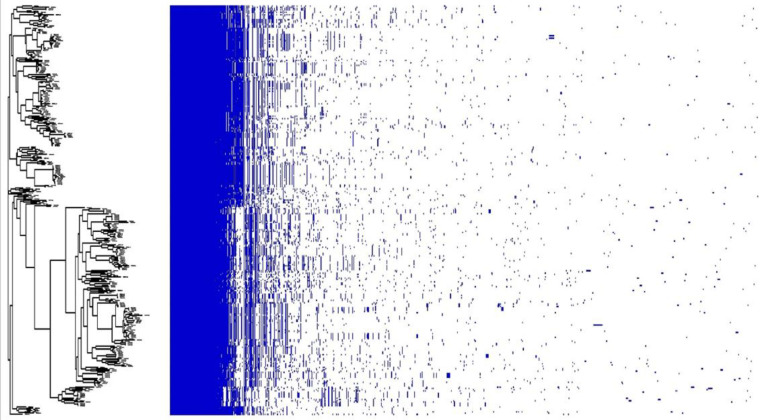
Phylogenetic tree generated based on presence and type of accessory genes in isolates and corresponding heat map representing absence or presence of genes in isolates.

**Figure 5 microorganisms-13-00907-f005:**
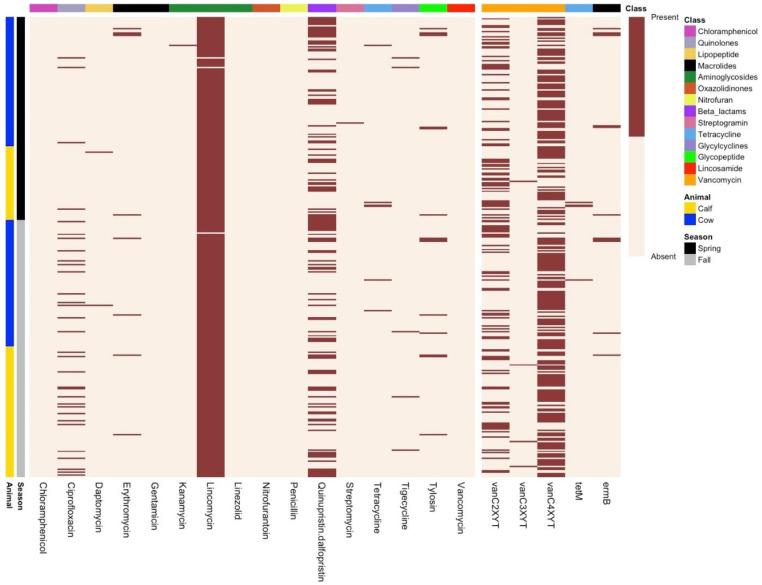
Antimicrobial resistance profiles of *Enterococcus casseliflavus* isolated and confirmed by whole genome sequencing from faeces collected from cows and calves in spring or fall (*n* = 331).

**Table 1 microorganisms-13-00907-t001:** Assembly statistics of 331 *E. casseliflavus* genomes isolated from cattle ^1^.

Assembly Stats	Average Value
N50 contig length (bp)	406,085
Number of contigs	35 (range: 8–114)
Number of contigs >= 1 kb	28
Number of contigs in N50	4
Genome size (bp)	3,674,039
GC content of contigs	42.4%

^1^ Sequence data for bacterial isolates are available at NCBI via BioProject PRJNA1191879.

## Data Availability

All Illumina sequence read data from the current study were deposited in the NCBI database as Short Read Archive (SRA) under BioProject ID PRJNA1191879.

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
