# Peer review of "Genomic Characterization of Enterococcus casseliflavus Isolated from Beef Cows and Calves"

_microorganisms, 2025, doi:10.3390/microorganisms13040907_

Round 1
Reviewer 1 Report (Previous Reviewer 3)
Comments and Suggestions for Authors
After reviewing the authors' responses, I reaffirm that the main concerns raised in the initial evaluation have not been adequately addressed. Therefore, my assessment remains unchanged for the following reasons:
1. Self-citations and contextualization: despite the authors' attempt to justify the high number of self-citations, the lack of external references limits the breadth and originality of the discussion. The manuscript requires a more comprehensive contextualization within the existing literature.
2. Originality and scientific contribution: while the authors argue that there is little information on E. casseliflavus in beef cattle, the study heavily relies on previous findings from the same research group. It does not provide significant innovation to justify publication in this journal.
3.External validation and generalization: the absence of additional controls and broader comparisons weakens the robustness of the conclusions. Although the effort to collect samples across a wide geographical area is acknowledged, it does not replace the need for greater external validation of the findings.
4. Relevance and impact: the justification provided does not change the assessment that the study has limited impact within the field and a narrow scope of interest. The lack of discussion on broader applications further diminishes its potential scientific contribution.
Given that the core issues from the initial review remain unresolved, the final recommendation is rejection of the manuscript.
Author Response
- The authors could reduce self-citations wherever possible, e.g. reference [1] could have a supporting reference.
As requested we have added a supporting reference “2” to the manuscript Manyi-Loh et al. 2018. We have also added some additional references related to the association of E. casseliflavus with clinical infections in humans.
- What is the full article/author list for reference [2]?
We have added the full reference for reference “2” which is now reference “3” due to the addition of support reference “2”.
- How were the genomes of the 331 strains separated during NovaSeq sequencing? Was each strain barcoded? Was each strain sequenced on a separate plate? The authors could provide more detail in the Methods section. This would also aid other researchers sequencing multiple strains (concurrently).
We have indicated that barcoded sequencing libraries were prepared for each lot of genomic DNA. This enabled us to relate the sequence data back to each of the individual isolates that were sequenced. We have also indicated the sequencing provider that undertook the sequencing on our behalf. Genome Quebec sequencing services (Montreal, PQ). We have also indicated that the sequence reads were de-multiplexed to enable assembly of each individual genome.
After reviewing the authors' responses, I reaffirm that the main concerns raised in the initial evaluation have not been adequately addressed. Therefore, my assessment remains unchanged for the following reasons:
- Self-citations and contextualization: despite the authors' attempt to justify the high number of self-citations, the lack of external references limits the breadth and originality of the discussion. The manuscript requires a more comprehensive contextualization within the existing literature.
We are quite familiar with the literature on enterococci – having just published a comprehensive review entitled “Enterococci as a One Health indicator of antimicrobial resistance.” By Zadi et al. Can. J. Micro. dx.doi.org/10.1139/cjm-2024-0024. In this extensive literature review, we only identified three references that related to bacteremia in humans as a result of E. casseliflavus. One of these references was already included in the manuscript – and we have added the other two to expand the context.
- Originality and scientific contribution: while the authors argue that there is little information on E. casseliflavus in beef cattle, the study heavily relies on previous findings from the same research group. It does not provide significant innovation to justify publication in this journal.
We see nothing wrong with employing previously used methodology developed in our laboratory to address new questions such as an exploration of the genomic nature of E. casseliflavus.
This is not an uncommon occurrence in science labs. A recent search of NCBI indicated that there are 276 E. casseliflavus genomes reported – with almost none of these being associated with published manuscripts. Of those that are associated with a published manuscript – none were isolated from beef cattle – and were almost exclusively isolated from humans. Consequently, our work with the addition of 331 E. casseliflavus genomes more than doubles the number of genomes that are currently available in the NCBI database. In our mind this contribution represents sufficient novelty, coupled with the fact that this is the first study that has directly addressed the genomics of E. casseliflavus isolated from cow herds.
- External validation and generalization: the absence of additional controls and broader comparisons weakens the robustness of the conclusions. Although the effort to collect samples across a wide geographical area is acknowledged, it does not replace the need for greater external validation of the findings.
We are uncertain with regard to what specific controls the reviewer is proposing that we include and unfortunately they offer no examples for us to consider. How does one identify a control in what is essentially an epidemiological study of beef cattle herds? Is the reviewer proposing that we inoculate animals? How do we know which cattle will possess E. casseliflavus before we actually try to isolate it? Sampling other animals in the vicinity of these farms such as deer or rodents would also make little sense.
- Relevance and impact: the justification provided does not change the assessment that the study has limited impact within the field and a narrow scope of interest. The lack of discussion on broader applications further diminishes its potential scientific contribution.
As this work essentially more than doubles the available E. casseliflavus genomes and is the first to focus on the genomics of this bacterium in an agricultural setting, we believe that it does have relevance and scope. There is a great deal of interest in using enterococci as an indicator of antimicrobial resistance in beef cattle and if an alternative species (E. casseliflavus) has implications for tracking it in the cow-calf sector vs E. hirae for tracking AMR in confined feedlot cattle – such a differentiation is of interest. We have also cited the available literature produce by other with regard to isolation of this species from cattle – which is sparse as we previously indicated.
Given that the core issues from the initial review remain unresolved, the final recommendation is rejection of the manuscript.
As per the reasons outlined above we disagree with this final conclusion.
Reviewer 2 Report (Previous Reviewer 1)
Comments and Suggestions for Authors
-
The introduction is comprehensive and provides relevant context on Enterococcus casseliflavus, particularly its role in One Health and antimicrobial resistance surveillance
-
The research design is well thought out, covering both phenotypic and genotypic characterization across a large number of isolates from multiple locations and seasons
-
Methodological details are clear and reproducible, including DNA extraction, sequencing, and analysis pipelines
-
The results are clearly presented with appropriate use of tables and figures to support findings
-
The discussion effectively contextualizes the findings within the broader literature and offers reasonable interpretations
-
The identification of resistance without known genetic determinants is intriguing and suggests potential for novel resistance mechanisms—this could be highlighted more explicitly
-
The seasonal variation in ciprofloxacin resistance warrants further exploration or hypothesis
-
The manuscript could benefit from minor language polishing in a few places to improve flow and readability
-
Consider discussing potential implications for monitoring E. casseliflavus in future AMR surveillance strategies in more depth
Author Response
-
The introduction is comprehensive and provides relevant context on Enterococcus casseliflavus, particularly its role in One Health and antimicrobial resistance surveillance
Thanks for the kind comment.
-
The research design is well thought out, covering both phenotypic and genotypic characterization across a large number of isolates from multiple locations and seasonsThanks for the kind comment.
-
Methodological details are clear and reproducible, including DNA extraction, sequencing, and analysis pipelines
Thanks for the kind comment
-
The results are clearly presented with appropriate use of tables and figures to support findings
Thanks for the kind comment.
-
The discussion effectively contextualizes the findings within the broader literature and offers reasonable interpretations
Thanks for the kind comment and we would note how this comment contrasts with the opinion of the other reviewer.
-
The identification of resistance without known genetic determinants is intriguing and suggests potential for novel resistance mechanisms—this could be highlighted more explicitly
We agree and have added a statement: "Further research to define the genetic factors responsible for Q/D resistance in E. casseliflavus is warrented." However, we believe that such an undertaking is beyond the scope of the current study.
-
The seasonal variation in ciprofloxacin resistance warrants further exploration or hypothesis
As we indicate - we have no viable explanation for the greater ciprofloxacin resistance in E. casseliflavus isolated from cows and calves in the fall vs those in the spring. We have indicated that this is the case in the manuscript. We have also added in the statement "This result was especially surprising given the infrequent use of fluoroquinolones within cow-calf herds [40].
-
The manuscript could benefit from minor language polishing in a few places to improve flow and readabilityWe have corrected wording where appropriate.
-
Consider discussing potential implications for monitoring E. casseliflavus in future AMR surveillance strategies in more depth
We have added a sentence to expand on this point: To summarize, E. casseliflavus is a prominent enterococcal species with the cow-calf sector, exhibiting notable genomic diversity among isolates. "Consequently, it may be a more reliable indicator of AMR in cow-herds than E. hirae, which tends to be the predominant species in feedlot cattle. "
Round 2
Reviewer 1 Report (Previous Reviewer 3)
Comments and Suggestions for Authors
There are still 2 self-citations, references 7 and 14, both cited in the introduction and number 14 in the discussion, but referring to Enterococcus hirae, which I as a reviewer would not like to be cited. Our reviews exist so that other authors can cite us and see the quality of our work and not ourselves. This practice of self-citation should be abolished from scientific writing, even though I know that journals allow a rate of up to 10-15% of self-citation.
I leave the final decision to the editor, based not only on my comments but also on those of the other reviewers. The authors have given "valid" answers, although not technical, and this is already the 3rd review of this manuscript and the academic editor must have the necessary elements to accept or reject the manuscript. I do not care if it is accepted, the only issue is and will be the unnecessary self-citation! I do not care if it is accepted, the only point is and will be the unnecessary self-citation! If the authors remove their self-citations, nothing in the text will be changed and if this is done I will give a favorable opinion, for now I will put accept after minor revisions.
Best regards
Author Response
We apologize for the over self-citation. We have deleted reference "7" and "14" as requested.
This manuscript is a resubmission of an earlier submission. The following is a list of the peer review reports and author responses from that submission.
Round 1
Reviewer 1 Report
Comments and Suggestions for Authors
The paper demonstrates significant strengths, including its comprehensive genomic and phenotypic characterization of Enterococcus casseliflavus isolates from beef cows and calves, which fills a critical gap in AMR studies for the cow-calf sector. The use of advanced whole genome sequencing and robust bioinformatics tools ensures high-quality data analysis, revealing the genetic diversity and open nature of the species' pangenome. By identifying intrinsic vancomycin resistance via the vanC gene cluster and its chromosomal localization, the study minimizes concerns over horizontal gene transfer. Its integration of a One Health approach effectively connects agricultural practices to antimicrobial resistance monitoring, contributing to global AMR management efforts. Additionally, the research highlights gaps in understanding specific resistance mechanisms, such as those for quinupristin/dalfopristin, setting the stage for future investigations.
The following shortcomings have been identified:
Limited Sample Size: The study analyzes only 20 Enterococcus casseliflavus isolates, which may not adequately represent the broader population in beef cows and calves, potentially limiting the generalizability of the findings.
Lack of Temporal Analysis: The research does not account for temporal variations in antimicrobial resistance patterns, missing potential seasonal or temporal trends that could influence resistance profiles.
Absence of Comparative Analysis with Other Enterococcus Species: Focusing solely on E. casseliflavus without comparing it to other Enterococcus species prevalent in bovine sources limits the contextual understanding of its resistance mechanisms and prevalence.
Insufficient Exploration of Resistance Mechanisms: While intrinsic vancomycin resistance is addressed, the study does not delve deeply into the genetic basis for resistance to other antibiotics, such as quinupristin/dalfopristin, leaving gaps in understanding the resistance mechanisms.
Limited Geographical Scope: The isolates are sourced from a specific region, which may not capture geographical variations in antimicrobial resistance patterns, thereby restricting the applicability of the results to other regions.
Potential Bias in Isolate Selection: The criteria for selecting isolates are not clearly defined, raising concerns about potential selection bias that could affect the study's outcomes.
Lack of Functional Validation: The study relies heavily on genomic data without functional assays to validate the predicted resistance genes, which could lead to inaccuracies in interpreting the resistance profiles.
Absence of Data on Horizontal Gene Transfer Potential: The research does not investigate the potential for horizontal gene transfer of resistance genes to other pathogenic bacteria, an important factor in understanding the spread of antimicrobial resistance.
Limited Discussion on Public Health Implications: The study does not sufficiently address the public health implications of finding antimicrobial-resistant E. casseliflavus in beef cattle, missing an opportunity to connect the findings to broader health concerns.
Omission of Virulence Factor Analysis: There is no assessment of virulence factors in the isolates, which is crucial for understanding the pathogenic potential of the strains studied.
Reviewer 2 Report
Comments and Suggestions for Authors
Dear authors,
I would like to offer some suggestions that I believe would help improve the manuscript.
The main comment is inconsistencies in the information regarding the presence or absence of resistance genes and phenotypic resistance in the results and discussion sections. The authors do not make any comments on this matter.
1. For instance, it would be interesting to discuss why phenotypic resistance to ciprofloxacin has been detected (“ciprofloxacin (9.6%, 32/331)” (line 227)), while no resistance genes (parC and gyrA) have been identified (lines 192-193). It is plausible that E. casseliflavus strains possess an efflux pump-mediated resistance mechanism. Studies with E. faecium have been conducted on this topic (doi: 10.3855/jidc.17304). Therefore, the authors should, at the very least, discuss the potential for this resistance mechanism to exist in E. casseliflavus, or better yet, test it through experiments involving efflux inhibitors.
2. In “3.2. Genomic characterization” of E. casseliflavus authors specify that vancomycin resistance determinant vanC-type locus (vanC-vanXY-vanT) was present in all of E. casseliflavus genomes (lines 185-190). At the same time in “3.3. Phenotypic resistance profiling” authors specify: “None of the isolates exhibited resistance to...vancomycin…” (lines 230-231).
How this disagreement can be explained? Please discuss in the text.
3. Please include the Table with antimicrobial susceptibility testing results.
4. Lines 278-280: “In contrast, no tetracycline resistance genes were found in E. casseliflavus and the macrolide resistance gene ermB was only rarely detected.” However, in “3.2. Genomic characterization of E. casseliflavus” there is no information available regarding tetracycline resistance genes other than that their presence has been indicated in Figure 4. At the same time in “3.3. Phenotypic resistance profiling” (lines 226-228) authors state that “Most isolates were resistant to…tetracycline (1.8%, 6/331)…”.
How this disagreement can be explained? Please discuss in the text.
5. When the authors reconcile the data and their analysis, it may be necessary to revise the conclusions.
Reviewer 3 Report
Comments and Suggestions for Authors
Based on the parameters provided, here's the evaluation of the uploaded article:
-
Introduction: The introduction provides a general context but could be improved by including more diverse references and expanding on the implications of the findings for the broader field. The cited references heavily rely on the authors' previous works, which may reduce the perceived depth and balance of the context provided.
-
Research Design: The research design is appropriate for genomic characterization and antimicrobial resistance studies. However, more clarity on the rationale for selecting specific methods or comparisons could enhance its robustness. The study might also benefit from additional controls or cross-comparisons with other systems to strengthen the conclusions.
-
Methods: The methods are described in detail, meeting the standards of reproducibility.
-
Results: Results are clearly presented, supported by comprehensive tables, figures, and appropriate statistical analyses.
-
Conclusions: The conclusions are well-supported by the results, effectively summarizing the key findings and their implications.
-
Autocitations: The article demonstrates excessive autocitation. Many references are from the same group and topic area, suggesting limited originality or contribution beyond their prior works. This could indicate a lack of external validation or broader relevance, which diminishes the scientific impact.
-
Originality/Novelty: The novelty is low, as much of the work builds on previous findings by the same group without significant new insights.
-
Content Importance: Medium importance, as the study focuses on a niche topic without broader applications clearly articulated.
-
Presentation Quality: High. The manuscript is well-structured and visually clear, with excellent graphical representations.
-
Scientific Solidity: Medium. While technically sound, the over-reliance on autocitations and limited external validation reduce the perceived robustness.
-
Reader Interest: Low, given the limited novelty and niche focus.
-
General Merit: Low. The manuscript does not make a substantial enough contribution to warrant publication in its current form.
General Recommendation: Rejection due to the lack of novelty, excessive autocitation, and limited broader significance. Suggestions for improvement include expanding the discussion to include external perspectives, reducing reliance on self-citations, and clearly articulating the study's implications for the field. Additional experiments or datasets that validate findings across diverse conditions could also enhance the manuscript.